# Centroid of the bacterial growth curves: a metric to assess phage efficiency
Nava Hosseini [1,2] ✉, Mahdi Chehreghani [3], Sylvain Moineau [1,2,4,5] & Steve J. Charette [1,2,6] ✉

Phage replication can be studied using various approaches, including measuring the optical density (OD) of a bacterial culture in a liquid medium in the presence of phages. A few quantitative methods are available to measure and compare the efficiency of phages by using a single index based on the analysis of OD curves. However, these methods are not always applicable to non-canonical OD curves. Using the concept of center of area (centroid), we developed a metric called Centroid Index (CI), sensitive to the trend of the growth curves (OD distribution) including bacterial regrowth, which is not considered by the methods already available. We also provide a user-friendly software to facilitate the calculation of CI. This method offers an alternative and more precise way to determine phage efficiency by considering the OD variations over time, which may help in the selection of phages for biocontrol applications.

Bacteriophages (or phages) are the most prevalent biological entities found in the biosphere. Phage characterization has contributed to numerous advances in life sciences, including molecular biology[1], biotechnology[2], and drug delivery[3]. The ability of virulent phages to efficiently kill specific pathogenic bacteria has made them an interesting option for therapeutic applications, which is often referred to as "phage therapy"[4]. They have been studied worldwide as a complement or alternative to antibiotics in many fields including medicine[5], aquaculture[6], agriculture[7], and veterinary sciences[8,9].

The efficacy of a specific phage to kill a target bacterium is usually first tested by a spot-on-lawn assay on solid medium[10]. Optical density (OD) measurements of bacterial growth in a liquid medium in the presence versus absence (uninfected control) of phages are also widely used[11]. Measuring the OD variations over time using a microplate spectrophotometer provides an opportunity to assess phage efficiency dynamics by comparing the growth of uninfected bacteria with phage-treated bacteria under desired incubation time and temperature[12]. One of the noticeable events that often occurs in phage-bacteria studies is the eventual regrowth of the bacterial population in the presence of phages. This phenomenon is often perceived as one of the obstacles in phage therapy. It is also a factor that needs to be considered in the selection of virulent phages for biocontrol or therapeutic applications[13].

To assess phage virulence, some authors have suggested using a common quantitative method[14–16]. Quantifying phage virulence facilitates the comparison of the lytic efficiency of various phages or phage cocktails against specific bacteria. This is achieved by providing an index value, in contrast to visual qualitative approaches, which may be inconclusive[16,17].

Among available quantitative methods for studying bacterial growth curves, the Virulence Index (VI) stands out by mainly focusing on the average behavior of the system through the calculation of the area under the curve. More specifically, regardless of the detailed trend of the growth curves, the ratio between the area under the control-bacterial curve and the area under the phage-infected bacterial curve is considered. One challenge when using this method to calculate the area under the curve is the selection of an appropriate end time that corresponds to the last data point in the experiments[14,15]. The end time should be long enough so that the steady state is approached, if any. When dealing with non-canonical growth curves, however, the upper bound (end time) is usually a pre-determined time which depends on many factors, including the purpose of the experiment, the bacterial strain, and the logistics of the experiment such as the spectrophotometer device[18]. An example of non-canonical growth curves, in which the steady state may not be approached, is when there is an increase in the OD linked to the bacterial regrowth. Other examples are variations in OD halfway through an experiment in the condition where phages were included as we observed in a previous study[17], or the OD decrease of the control conditions, particularly in experiments with a long incubation period[17].

The maximum specific growth curve, known as $\mu_{max}$, is another approach[19] for the interpretation of the bacterial growth curves. However, it

[1]Institut de Biologie Intégrative et des Systèmes (IBIS), Pavillon Charles-Eugène-Marchand, Université Laval, Quebec City, QC G1V 0A6, Canada. [2]Département de biochimie, de microbiologie et de bio-informatique, Faculté des sciences et de génie, Université Laval, Quebec City, QC G1V 0A6, Canada. [3]Department of Mechanical Engineering, McGill University, Montreal, QC H3A 0C3, Canada. [4]Groupe de Recherche en Écologie Buccale (GREB), Faculté de médecine dentaire, Université Laval, Quebec City, QC G1V 0A6, Canada. [5]Félix d'Hérelle Reference Center for Bacterial Viruses, Université Laval, Quebec City, QC G1V 0A6, Canada. [6]Centre de Recherche de l'Institut Universitaire de Cardiologie et de Pneumologie de Québec (IUCPQ), Quebec City, QC G1V 4G5, Canada. ✉e-mail: nava.hosseini.1@ulaval.ca; steve.charette@bcm.ulaval.ca

may also be challenging, especially for non-canonical growth curves, with different growth levels ($N_{\text{asymptote}}$), lags, and non-exponential growth patterns[14,18]. $N_{\text{asymptote}}$ or the peak growth is another parameter that was previously suggested to be included in bacterial growth curve analysis[14]. Using the $N_{\text{asymptote}}$ value in phage virulence analysis is not always helpful since reaching a discernible stationary phase might not always be possible, especially in the case of bacterial regrowth[18].

Any method involving curve fitting, such as the phage score[15], assumes that bacterial growth curves follow a canonical growth pattern. These methods also require the implementation of a numerical approach. For instance, the phage score method uses the logistic equation for bacteria and the generalized double exponential logistic curve for phage-treated bacteria[15]. However, the dynamics of a biological system may not always follow a unique mathematical model[20]. The parameters involved in the mathematical expression of the dynamics of the system obtained via the curve fitting technique are not unique, and this lack of uniqueness may limit the applicability of the method[18]. Also, an imprecise fit of the model due to numerical errors may create a discrepancy between the experimental and fitted data. Despite the advantages of analytical integration, the errors mentioned might be more pronounced than the errors resulting from the use of trapezoidal integration in the VI method. As the obtained analytical expression via this method is itself generated by a numerical approach in the previous steps (fitting the experimental data on a logistic or generalized double exponential logistic equation using a numerical approach), the inherent errors may propagate to the analytical integration.

The limitations of the current tools suggest that an alternative approach is necessary. Here, using the concept of the center of area, or centroid, which is the arithmetic mean position of a body[21], we propose a quantitative approach for phage virulence interpretation. Specifically, the proposed method is sensitive to the trend of the growth curves during phage-bacteria interactions, as the distribution of OD values during incubation time is considered. We also discuss circumstances in which using the proposed method grades better the efficiency of phage cocktails, as compared to other available methodologies.

## Results and discussion
### Development of the mathematical expression for CI
The centroid of a two-dimensional or three-dimensional shape, referred to as the geometric center or center of the figure, represents the average location of all the points within the surface of the shape. In mathematics or physics, the center of gravity, is the specific location where the combined effect of an object's weight produces no net moment. This equilibrium condition would allow a rigid body to remain balanced on the tip of a finger positioned at its center of gravity. Suppose a homogeneous flat plate with a uniform thickness, then the center of gravity, denoted as point $G$ in Fig. 1a, would coincide with the centroid of the plate, shown in Fig. 1b.

In a two-dimensional plane, the centroid of a bounded region $A$ in the $xy$-plane, shown in Fig. 1b, is defined as follows[21]:

$$\bar{x} = \frac{Q_y}{A}, \bar{y} = \frac{Q_x}{A}, \tag{1}$$

where $A = \iint_A dA$ is the area of the region $A$, $dA$ is a small rectangle of sides $dx$ and $dy$, and $Q_y = \iint_A x \, dA$ and $Q_x = \iint_A y \, dA$ are known as the first moments of area $A$ with respect to the $y$- and $x$-axis, respectively. The first moment of area is associated with the topological distribution of an area with respect to an axis. With respect to a centroidal axis, the first moment of area vanishes.

Evaluation of the integrals associated with Eq. (1) requires a double integration with respect to $x$ and $y$. However, usually, it is possible to choose $dA$ to be a strip element rather than a rectangle of sides $dx$ and $dy$, as shown in Fig. 2. As a result, the problem can be further simplified, enabling the performance of a single integration instead of a double integration. Denoting the centroid of each strip by $\bar{x}_j$ and $\bar{y}_j$, the coordinates of the centroid of area $A$ can be expressed as follows:

$$\bar{x} = \frac{\int \bar{x}_j dA}{A}, \bar{y} = \frac{\int \bar{y}_j dA}{A}. \tag{2}$$

In the case of our optical density measurements, the data were collected every 15 min (sampling time). This results in $n$ strips, where $n$ is the number of time intervals. Each strip has sides of "sampling time" and measured "OD values." Therefore, one can use the discretized form of the integrals of Eq. (2), yielding:

$$\bar{x} = \frac{\sum_{j=1}^n \bar{x}_j A_j}{A}, \bar{y} = \frac{\sum_{j=1}^n \bar{y}_j A_j}{A}, \tag{3}$$

where $A = \sum_{j=1}^n A_j$, in which $A_j$, the area of each of the strips, can be expressed as:

$$A_j = \frac{y_{j-1} + y_j}{2}(x_j - x_{j-1}), \tag{4}$$

and $\bar{x}_j$ and $\bar{y}_j$ are the coordinates of the centroid of the $j$th strip ($j$th time interval), as in Fig. 2a, that for the chosen trapezoidal strips are:

$$\bar{x}_j = x_{j-1} + \frac{(x_j - x_{j-1})(y_{j-1} + 2y_j)}{3(y_{j-1} + y_j)}, \bar{y}_j = \frac{1}{3}\left(y_j + \frac{y_{j-1}^2}{y_{j-1} + y_j}\right). \tag{5}$$

Let $(\bar{x}_{\text{ctrl}}, \bar{y}_{\text{ctrl}})$ denote the centroid of the region enclosed by the curve of the control bacteria and the horizontal axis. Also, let $(\bar{x}_i, \bar{y}_i)$ represent the

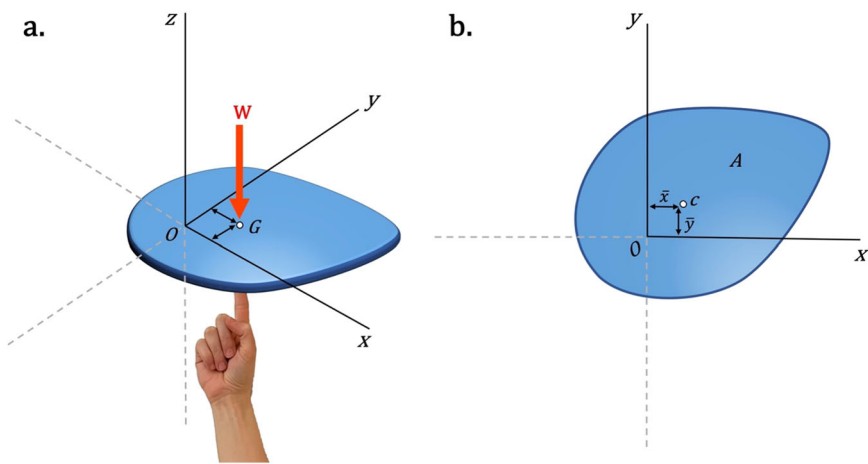

**Fig. 1 | The center of gravity and centroid of a homogeneous flat plate. a** The schematic representation of the "center of gravity," point $G$, for a plate. The combined effect of the weight of the plate produces no net moment at the center of gravity. Hence, one can assume that the total weight of the body, shown by vector **W** is applied at point $G$. **b** For a homogeneous flat plate, the center of gravity, $G$, coincides with the centroid of the flat plate, denoted by $c(\bar{x}, \bar{y})$. "$A$" stands for the area of the flat plate.

**Fig. 2 | Representation of the centroid. a** A view of the centroid of the *j*th trapezoidal strip (in red) with an area $A_j$ in a putative growth curve (dark blue curve). The *x*-axis corresponds to time and the *y*-axis represents OD. $\bar{x}_i$ and $\bar{y}_i$ are the coordinates of the centroid of the *j*th strip. **b** The schematic representation of the centroid of a bacterial curve and the phage i-infected bacterial curve, in which the centroids are shown by colored diamonds.

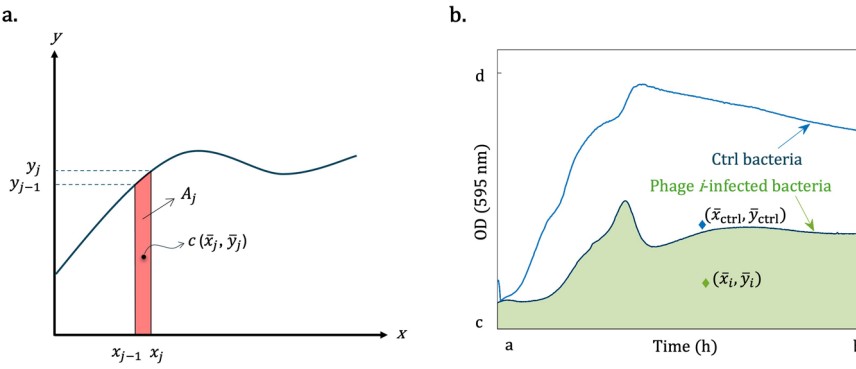

**Fig. 3 | Uncertainty of asymptotic growth ($N_{asymptote}$) prediction when the steady state has not been approached during the experiment.** Here, biological data displaying non-canonical curves are shown[17]. Even after 96 h of phage-bacteria incubation, the determination of the $N_{asymptote}$ is not possible because a stationary phase or a maximum growth level has not been obtained (any condition such as A, B, or C could have occurred if the experiment had been continued for more than 96 h). In this situation, the prediction of $N_{asymptote}$ is based on regression, which may not be a robust procedure due to the complexity of the system.

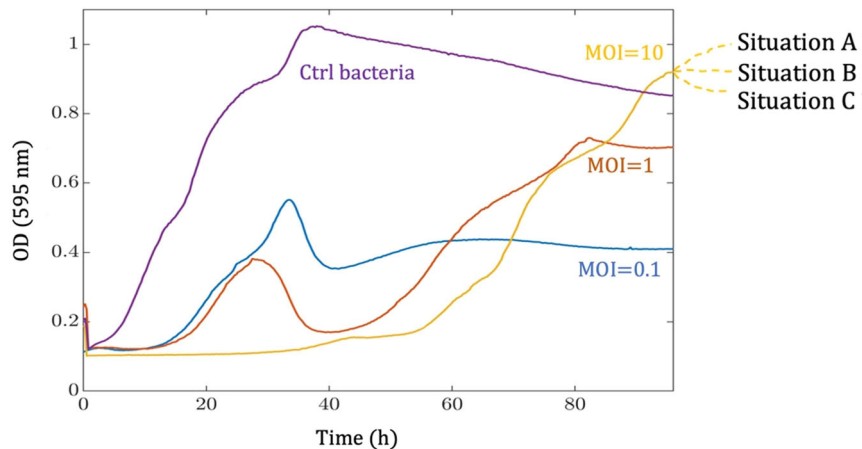

centroid of the region enclosed by the growth curve of bacteria infected by the *i*th bacteriophage and the horizontal axis, as shown in Fig. 2b. We define the centroid index for the bacteriophage, $\mathrm{CI}_i$, as follows:

$$\mathrm{CI}_i = 1 - \frac{\bar{x}_i \bar{y}_i}{\bar{x}_{ctrl} \bar{y}_{ctrl}}. \tag{6}$$

Note that the centroids of the growth curves can be calculated using Eq. (3). To facilitate the calculation of the CI of any bacterial curve, we provide a user-friendly software, as shown in Supplementary Fig. 1 and explained in Supplementary Note 1. The value of CI ranges from 0 to 1 in which a CI of 1 indicates an efficient bacteriophage in lysing a given bacterial strain, whereas a CI of 0 specifies an inefficient one. This is because an ideal phage or a phage cocktail would have a centroid with both a small $\bar{x}_i$ and $\bar{y}_i$, thereby, neither having a great value of bacteria OD, nor phage resistance towards the end of incubation (typically demonstrated by the late growth of bacteria). Indeed, an effective phage for therapeutic purposes should yield a condition where the bacteria population is decreased, and without bacterial regrowth. In some cases, when the ratio of $\frac{\bar{x}_i \bar{y}_i}{\bar{x}_{ctrl} \bar{y}_{ctrl}}$ exceeds 1, the CI value becomes negative. This may happen when the OD of phage-treated bacteria grow even higher than the control bacteria ($\bar{y}_i > \bar{y}_{ctrl}$), or a substantial growth has happened towards the end of incubation time ($\bar{x}_i > \bar{x}_{ctrl}$) or both. This phenomenon may be observed in the laboratory when a notable amount of phage lysate is used, favoring the growth of phage-resistant cells[22]. Therefore, the negative values of CI may demonstrate that the phage is inefficient.

### Challenges associated with available approaches for phage virulence quantification

Methods such as $I_{sc}$[14] are based on the assumption that bacterial growth curves would eventually approach a stationary phase or a maximum growth

level. In fact, the biological data may not always respect this principle since the presence of phages may have an impact only for a certain period of time or have a partial effect without producing a canonical curve. Unless a distinct stationary phase or maximum growth level has been obtained during the experiment, the calculated asymptotic number or ratio would be an extrapolated value that is not representative of the biological situation. It is known that the dynamics of a complex system, where many interactions (and strong nonlinearities) are involved, is not always predictable far into the future[23].

In phage-bacteria interaction assays, for the untested situation where the future OD values have not been captured by the spectrophotometer, any situation may occur (Fig. 3). Competition between bacteria for limited food resources and the parallel interplay with phages are both involved in the subsequent destiny of such interaction. Therefore, methods that attempt to predict the maximum growth levels ($N_{asymptote}$) through regression for the phage-treated bacteria may not be optimal. Additionally, the regression process itself can be unstable, with different initial values leading to different growth parameters or failure to converge during iterations. In such cases, the chosen model can greatly influence the magnitude of the calculated asymptotic level. Different statistical fit criteria can also produce divergent results due to the presence of outliers[18].

In the phage score method[15], it is assumed that bacterial growth would follow the logistic growth model and the phage-treated bacteria can be modeled using a generalized double exponential logistic equation. Then, a curve fitting procedure is performed, resulting in mathematical expressions for the growth curves. The obtained expressions can be used to estimate the area under the growth curves by performing analytical rather than numerical integration. However, growth curves may not fall into this canonical model, as shown in Fig. 3. Additionally, the curve fitting process inherently contains numerical errors. The existence of outliers, which are data points that can markedly deviate from the general pattern, accentuates

these errors, as the outliers have a substantial impact on the curve fitting process[18]. Therefore, even though phage score method avoids numerical errors due to trapezoidal integration, it introduces other errors that may be more substantial than performing a numerical integration.

Using $\mu_{max}$ or maximum specific growth rate[19], which is the slope of the growth curve at its exponential region for the estimation of the phage virulence, is also not sufficient for the calculation of the virulence[14,18]. For instance, as shown in Fig. 4, two putative phage-treated bacterial curves have the same maximum specific growth rates and therefore, this method fails to distinguish between them.

The VI is one of the most frequently used methods to quantify phage virulence. It is based on the comparison of the area under the phage-treated and control-bacterial growth curves. There are some disadvantages to this method, as discussed previously by others[14,15]; one is that it considers the area regardless of the curve trend (increasing or decreasing). In other words, the VI is not able to discriminate whether a given phage inhibit the bacterial growth at the beginning of the incubation period or at the end. Hence, the same area with different curve trends might lead to the same index, as shown in Fig. 5. The CI, on the other hand, considers the variation of OD points versus time. Therefore, CI can distinguish between the two growth curves with identical areas under the curve but with different trends. As shown in Fig. 5, since phage A and phage B have the same mean value (area under the curve), VI predicts the same index for both phages, but CI indicates that phage B is more efficient.

## Biological significance of CI

We can conceptualize a bacterial growth curve in geometric terms for analytical purposes since it shows geometrical properties. In a bacterial growth curve, we typically plot time on the $x$-axis and the optical density, representing the population density, on the $y$-axis. This creates a curve that represents the change in bacterial population over time. The bacterial growth curve itself has a shape that changes over time, starting with a lag phase, followed by an exponential growth, then reaching a plateau during the stationary phase, and potentially declining during the death phase. We can think of the centroid as a hypothetical point that represents the "center of mass" or "balance point" of the growth curve. It is not a physical point within the curve, but a conceptual one, that summarizes the distribution of the bacterial population densities over time. In other words, the position of the centroid can indicate where the growth is centered on the bacterial growth curve, analogous to how it represents the balance point of a geometric object (center of gravity of a physical object). Changes in the position of the centroid indicate shifts in the distribution of cell population densities over time. This particularly facilitates the comparison between the bacterial growth affected by a unique phage or a combination of phages, since each growth curve may exhibit unique trends due to a variety of reasons (type of phages, mutations in bacterial host, etc.).

The centroid of any growth curve, having two coordinates ($\bar{x}$ and $\bar{y}$) provides information regarding:

1. The magnitude of cell density or growth ($\bar{y}$)
2. The timing of when this density occurs ($\bar{x}$ or $\bar{t}$).

For example, $\bar{t}$ shifting towards later times might indicate the regrowth of the bacterial population, and the upward $\bar{y}$ might represent a higher magnitude of cell density or the bacterial regrowth. So far, there are no other methodologies which pinpoint where the cell density/growth is located, and the regrowth of the bacterial population is mostly not adequately or not at all considered by current indices (see Fig. 6a). Overall, by considering the locations and timings of the peaks, the centroid index (CI) provides additional information regarding the phage-bacteria dynamics over time.

### Implementation of CI on biological data

In a previous study, we tested a four-phage cocktail at different Multiplication of Infection (MOI) on two different strains of *Aeromonas salmonicida* subsp. *salmonicida*[17]. This bacterial species is psychrophilic, with the optimal growth around 20 °C[24]. Given its relatively slow growth, even slower than other mesophilic *A. salmonicida* subspecies at this

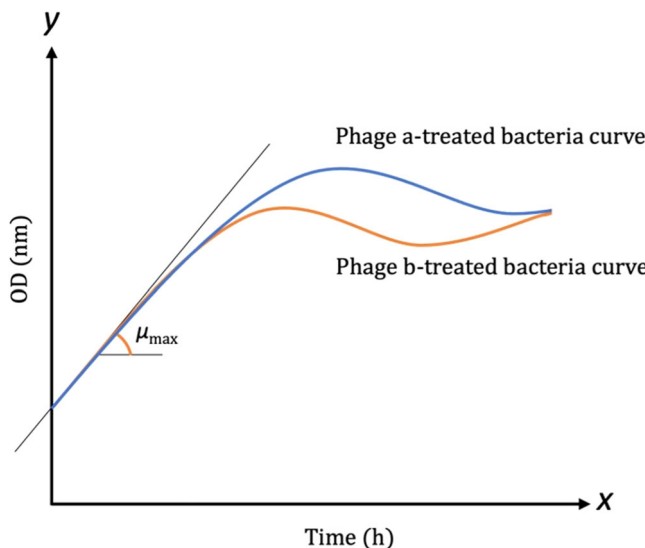

**Fig. 4 | The schematic representation of the putative curves for the comparison of $\mu_{max}$ (maximum specific growth rate) when there are two similar slopes for the exponential region of the two phage-treated bacterial curves (Phage a and Phage b).** Since the same slopes are obtained, using $\mu_{max}$ is not a sufficient parameter for the virulence comparison.

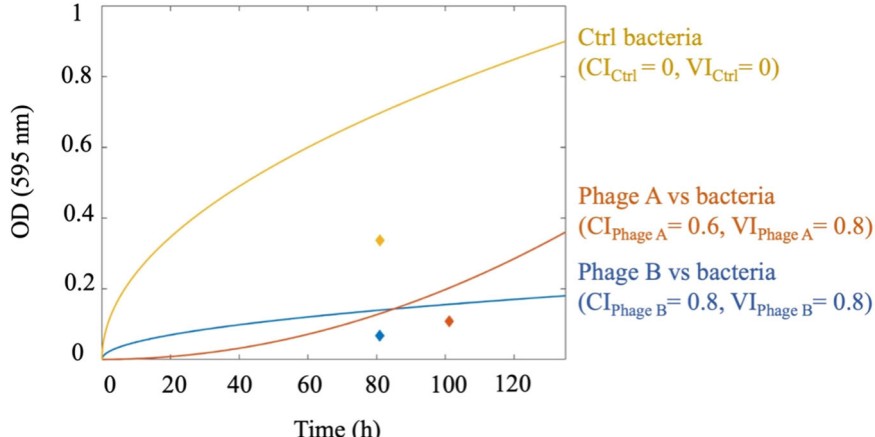

**Fig. 5 | The comparison of the virulence index (VI) method with the centroid index (CI) method using two putative curves.** The centroid location of each curve is shown by diamonds colored to match the curve. The calculated VI and CI for each curve is shown by the same color. The variation of OD points over time affects the centroid value, but the VI method is only sensitive to the area under the curves.

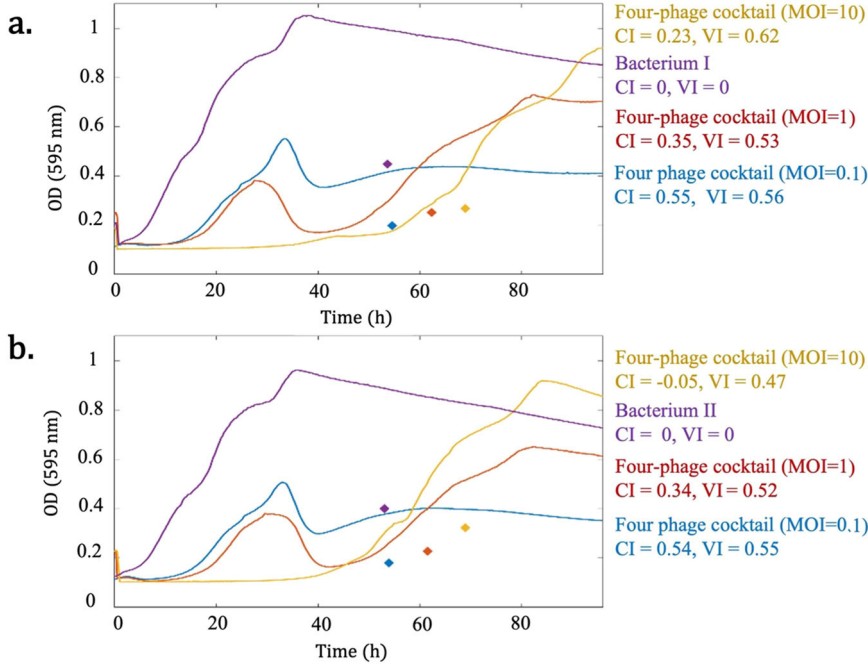

**Fig. 6 | The comparison of virulence index (VI) and centroid index (CI) calculations performed on the OD values of two biological examples obtained from ref. 17.** The centroid for each curve is shown by diamonds with the same color. **a** Bacterium I infected by a four-phage cocktail at three MOIs (0.1, 1, and 10). VI suggests that the four-phage cocktail at MOI = 10 is the best for lysing Bacterium I, whereas CI indicates that the four-phage cocktail at MOI = 0.1 is the best condition. This shows that VI does not consider the bacterial regrowth. **b** Bacterium II infected by the same four-phage cocktail at three MOIs (0.1, 1, and 10). Both VI and CI indicate that a four-phage cocktail at MOI = 0.1 is the most efficient condition. We conclude that CI is more robust, as it does not only rely on the area under the curves.

temperature, growth experiments were carried out over longer periods, 72 or 96 h[25,26]. We noticed that the use of VI to estimate phage virulence with our data was problematic. As seen in Fig. 6a, we tested the same four-phage cocktail at three MOI (0.1, 1, 10), which led to three phage-infected bacterial growth curves[17]. The highest calculated VI was obtained with the MOI of 10 (yellow curve). Yet, a considerable increase in OD values was observed toward the end of the incubation time due to bacterial regrowth, which is considered a detrimental outcome for phage therapy[27]. Therefore, obtaining the highest VI value for such a condition is not biologically accurate. This situation suggests the need for modifying the phage cocktail or reevaluating the initial MOI. In Fig. 6b, the condition with the highest VI value, or the most efficient cocktail, is associated with the four-phage cocktail at MOI = 0.1 (blue curve). Focusing on the blue curves in both Figs. 6a and 6b, which both represent the four-phage cocktail at MOI = 0.1, one can notice that these conditions exhibit the lowest bacterial regrowth towards the end of incubation time, as compared to other conditions. Since the curves are somewhat biologically equivalent in both Figs. 6a and 6b, failing to assign the highest index to the two conditions with the highest bacterial lysis raises questions about the consistency of VI. In other words, these examples demonstrate that VI is not a robust method, as it only depends on the area under the curve regardless of the biological concept behind the phage-treated bacterial curves. On the other hand, according to the CI calculation, the four-phage cocktail at MOI = 0.1 obtained the highest score, as compared to other curves. It suggests that the prediction of virulence efficacy by CI is biologically more reliable.

Here, we also tested additional phage cocktails with related compositions to show that minor modifications, such as changing a single parameter (e.g., the presence of CaCl₂) or adding or removing phage(s) from the cocktail, could result in considerable variations in virulence behavior. These data illustrate the dynamics and complexity of the phage-bacteria interactions as phage-treated bacterial curves can be unpredictable.

The new phage cocktail data demonstrate that using either VI or CI can consistently predict the same cocktail as the most efficient option against the target bacteria A and B (Fig. 7). In Fig. 7a, b, the CI method ranks the phage cocktails in the same order as the VI method does. Thus, the VI measurement may accurately capture the characteristics of the curves based on their actual behavior in some cases (Fig. 7). However, there are instances where it either falls short or fails to do so (Bacteria C and D, Fig. 8).

In Fig. 8a, VI classified phage Cocktails 4 and 5 under the same phage virulence (to the second decimal place), while it is an incorrect estimation according to the pattern of these two curves. Cocktail 4 showed a considerable OD increase after 60 h of incubation, and it is after 90 h that it reached a somewhat steady pattern. In contrast, the emergence of bacterial regrowth towards the end of the incubation period was lower for Cocktail 5. This phenomenon is reflected on the CI value, indicating that the CI value can predict the efficiency of phages at a higher resolution. A similar phenomenon was observed in Fig. 8b, where both Cocktails 2 and 6 had the same VI, but the CI method was able to differentiate the two curves, giving a higher score to Cocktail 6. All OD data which were investigated in this study, in addition to the VI and CI calculations are available in Supplementary Data 1.

In addition to assessing the impact of phages on the psychrophilic bacterium like *A. salmonicida* using the centroid index, we broadened our assessment of the CI applicability by testing other bacterial species infected with phages. The first example is from a study on *Streptococcus mutans*[28]. In this study, authors assessed the VI values for phage SMHBZ8 against *S. mutans*, at six different MOIs (0.01, 0.1, 1, 10, 100, and 1000). The highest VI was with the MOI of 1000 (Fig. 9a). Assigning the highest VI value to this condition appears biologically plausible, since the bacterial growth inhibition occurred until at least 45 h post-incubation, with an increase of OD after this time, followed by a plateau in OD at about 55 h and until the end of incubation time. According to our evaluation, the CI assigned the same MOI as the most efficient condition. Thereby, in this example, both VI and CI identified the MOI of 1000 as the most efficient condition against *S. mutans*. However, for the other MOIs, the CI and VI classifications were not the same (for CI: 1000, 1, 0.1, 10, 100, 0.01, and for VI: 1000, 1, 10, 100, 0.1, 0.01). It appears that the curve for the MOI of 0.1 was not perceived in the same manner by the two analytical tools. Under the CI metric, an MOI of 0.1 ranked third, whereas under the VI metric, it ranked as the penultimate position. In this specific case, VI only highlighted the early bacterial growth observed between 10 and 25 h, although there was no further increase in OD after 25 h post-infection for this condition. For its part, CI allowed us to contextualize this plateau observed until the end of incubation time with a final OD similar to the MOIs of 1000 and 1 (Fig. 9a).

We examined another study conducted on *Burkholderia cenocepacia*[29]. In Fig. 9b, the virulence of podophage JC1 was investigated at six MOIs (0.001, 0.01, 0.1, 1, 10, 100, and 1000) against *B. cenocepacia* Van1 at 30 °C.

**Fig. 7 | Two biological examples where virulence index (VI) and centroid index (CI) rankings are equivalent for all phage cocktails. a** Bacterium A grown in the presence of six phage cocktails at MOI = 1. The centroid of each curve is shown by a diamond with the corresponding color. The calculated CI and VI are shown in front of the cocktail number. Each color corresponds to the assigned number of the cocktail. According to both the VI and CI, the most to least effective phage cocktails are 3, 5, 1, 6, 4, 2. **b** The same phage cocktails were tested with Bacterium B, and the CI and VI were calculated accordingly. For Bacterium B, the ranking of the cocktails from the best to worst is: 4, 2, 6, 5, 3, 1.

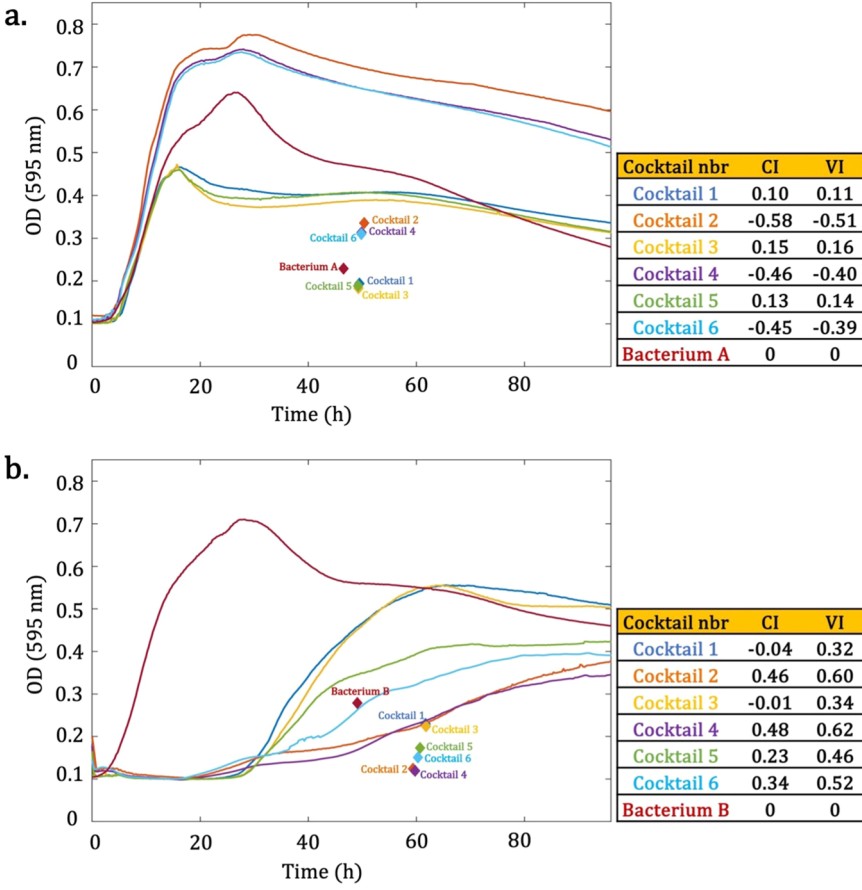

| Cocktail nbr | CI | VI |
|---|---|---|
| Cocktail 1 | 0.10 | 0.11 |
| Cocktail 2 | -0.58 | -0.51 |
| Cocktail 3 | 0.15 | 0.16 |
| Cocktail 4 | -0.46 | -0.40 |
| Cocktail 5 | 0.13 | 0.14 |
| Cocktail 6 | -0.45 | -0.39 |
| Bacterium A | 0 | 0 |

| Cocktail nbr | CI | VI |
|---|---|---|
| Cocktail 1 | -0.04 | 0.32 |
| Cocktail 2 | 0.46 | 0.60 |
| Cocktail 3 | -0.01 | 0.34 |
| Cocktail 4 | 0.48 | 0.62 |
| Cocktail 5 | 0.23 | 0.46 |
| Cocktail 6 | 0.34 | 0.52 |
| Bacterium B | 0 | 0 |

While the authors had previously determined the VI values for their growth curves, we re-evaluated the same curves using CI. As a result of comparison between VI and CI values, we observed that CI attributed higher indices to curves having lower regrowth after 45 h of incubation. In fact, CI assigned the MOI of 100 as the most efficient condition in Fig. 9b since the regrowth of bacteria was lower compared with other curves. However, the VI calculated by the authors identified the MOI of 1000 as the most efficient condition, which contradicts the biological interpretation of the curve, displaying an upward trend. This example clearly demonstrates that it is not the absolute amplitude of growth that should be considered, but the positioning of this growth over time. The tools developed in the past did not take this temporal aspect into consideration whereas the centroid index does, making the CI metric more accurate.

In summary, the existing virulence indices provide a means to standardize and quantify the interaction between phages and bacteria. These comparative tools can provide guidance in selection of potential phages in specific applications (phage therapy or biocontrol applications) or aiding in the screening and optimization of phage candidates[15,16]. However, virulence in multi-phage/single-host systems can involve highly dynamic fluctuations over time, influenced by a variety of factors such as phage-host coevolution, mutational resistance, counter-resistance[30,31], competition between phage variants during incubation time, etc[32]. The centroid index method considers these temporal fluctuations. The centroid index or any other available indices may not capture all the detailed fluctuations of a growth curve and therefore depending solely on these indices may overlook subtle nuances in phage behavior. Nevertheless, by presenting examples derived from our research as well as from existing literature, we have illustrated the robustness of the CI method in validating both canonical and non-canonical growth curves, underscoring the efficacy and reliability of this method, especially when a gradual increase in OD occurs after prolonged incubation.

## Conclusions

We suggest an alternative metric, named CI, to quantify the lytic efficiency of an individual phage or phage cocktail. Previously proposed methods were shown to have limitations, including assuming unrealistic mathematical models, numerical errors due to curve fitting, and neglecting the OD curve trend. Through biological data, it was demonstrated that the CI resulted in a more reliable and robust quantification of phage lytic activity, as it takes the distribution of the entire bacterial growth curve into account, including bacterial regrowth after extended incubation. Therefore, we proposed using the centroid index, which is both easy to implement and capable of overcoming the aforementioned limitations.

## Methods

### Phages, bacteria, phage cocktails, and growth curves

In this study, we used four strains of the Gram-negative slow-growing psychrophilic species *A. salmonicida* subsp. *salmonicida* namely, strain 01-B526 (identified hereafter as Bacterium A), 01-B516 (B), M22710-11 (C), and SHY18-4069 (D)[33–35]. We also used six previously characterized lytic phages, including five myophages (44RR2.8t.2, 65.2, Riv-10, SW69-9, SW69-9.BK93) and one podophage (MQM1)[17,36,37]. Six different phage cocktails were designed and used at a multiplicity of infection (MOI) of 1 (Table 1). Additional information on the phages and bacteria used are provided in Supplementary Table 1 and 2.

Phage cocktails and bacteria were incubated at a final volume of 300 µL at $OD_{595}$ of 0.1, in flat-bottom 48-well microplates for 96 h at 19 °C, using a microplate reader (Infinite 200 PRO, Tecan, Baldwin Park, CA, USA). The plates oscillated constantly with a 3.5 mm amplitude in orbital mode at 200 rpm, and optical density data were collected every 15 min at 595 nm[17]. The experiment was done in biological duplicates. Additionally, OD data obtained from our previous study[17] on the effect of a four-phage cocktail (Table 1) at three MOI (0.1, 1, and 10), on two *A. salmonicida* subsp.

**Fig. 8 | Two biological examples illustrate the superior interpretative validity of the centroid index (CI) over the virulence index (VI) in assessing phage activity, where VI yielded identical values for divergent curve trends.** The values for VI and CI are calculated for six phage cocktails tested against Bacterium C and D, respectively. **a** To the second decimal, VI calculated the same values for Cocktail 4 and Cocktail 5 (shown in red in the table), while CI differentiates these two cocktails. **b** Cocktail 2 obtained the same VI value as Cocktail 6 (shown in red in the table), while Cocktail 2 had an OD increase after 60 h of incubation, which did not drop until approximately 90 h. Meanwhile, Cocktail 6 had approximately reached a steady state after 60 h and did not have an OD increase after this incubation time.

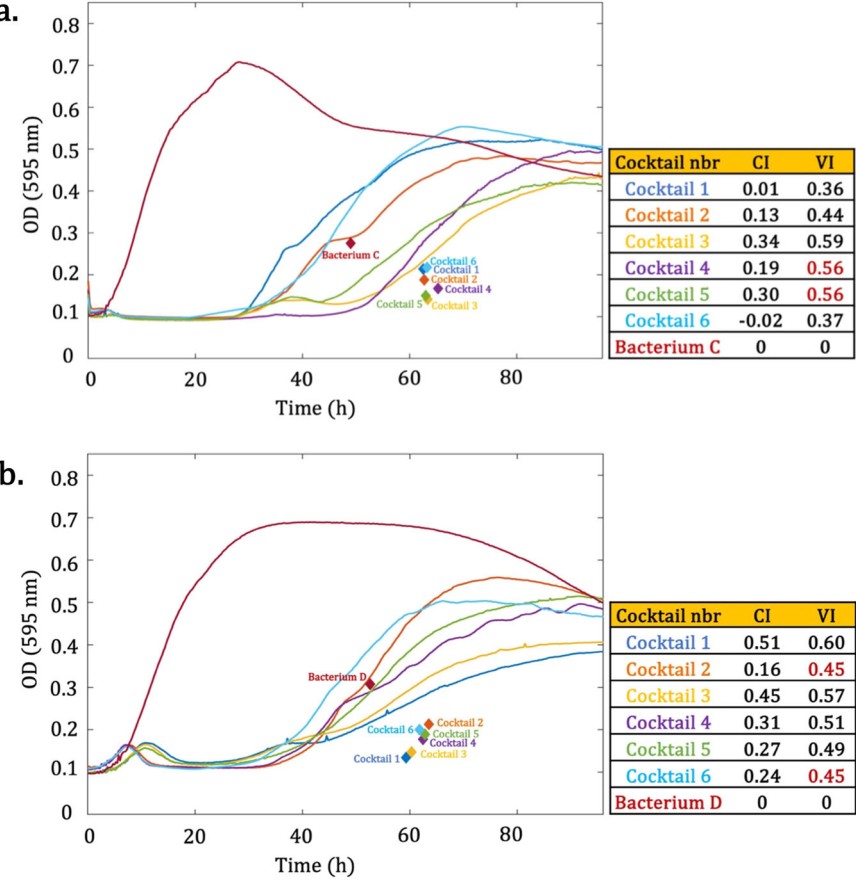

| Cocktail nbr | CI | VI |
|---|---|---|
| Cocktail 1 | 0.01 | 0.36 |
| Cocktail 2 | 0.13 | 0.44 |
| Cocktail 3 | 0.34 | 0.59 |
| Cocktail 4 | 0.19 | 0.56 |
| Cocktail 5 | 0.30 | 0.56 |
| Cocktail 6 | -0.02 | 0.37 |
| Bacterium C | 0 | 0 |

| Cocktail nbr | CI | VI |
|---|---|---|
| Cocktail 1 | 0.51 | 0.60 |
| Cocktail 2 | 0.16 | 0.45 |
| Cocktail 3 | 0.45 | 0.57 |
| Cocktail 4 | 0.31 | 0.51 |
| Cocktail 5 | 0.27 | 0.49 |
| Cocktail 6 | 0.24 | 0.45 |
| Bacterium D | 0 | 0 |

**Fig. 9 | The assessment of the applicability of the centroid index (CI) on other bacterial species.** OD values versus time were obtained by WebPlotDigitizer to re-draw the bacterial growth curves and to calculate the CI and VI values. **a** The virulence efficiency of phage SMHBZB on the growth of *S. mutans* at MOIs of (0.01, 0.1, 1, 10, 100, and 1000). OD values were plotted on a logarithmic scale. In this example, VI and CI take the MOI of 1000 as the most efficient condition against *S. mutans*. The original data were from ref. 28. **b** The virulence efficiency of phage JC1 on the growth of *B. cenocepacia* Van1, at MOIs of (0.001, 0.01, 0.1, 1, 10, 100, and 1000). While VI considered the MOI of 1000, showing a considerable bacterial regrowth/OD increase after 25 h as the most efficient condition for inhibiting the growth of Van1, the CI value suggested the MOI of 100 as the most efficient condition. The original data were from ref. 29.

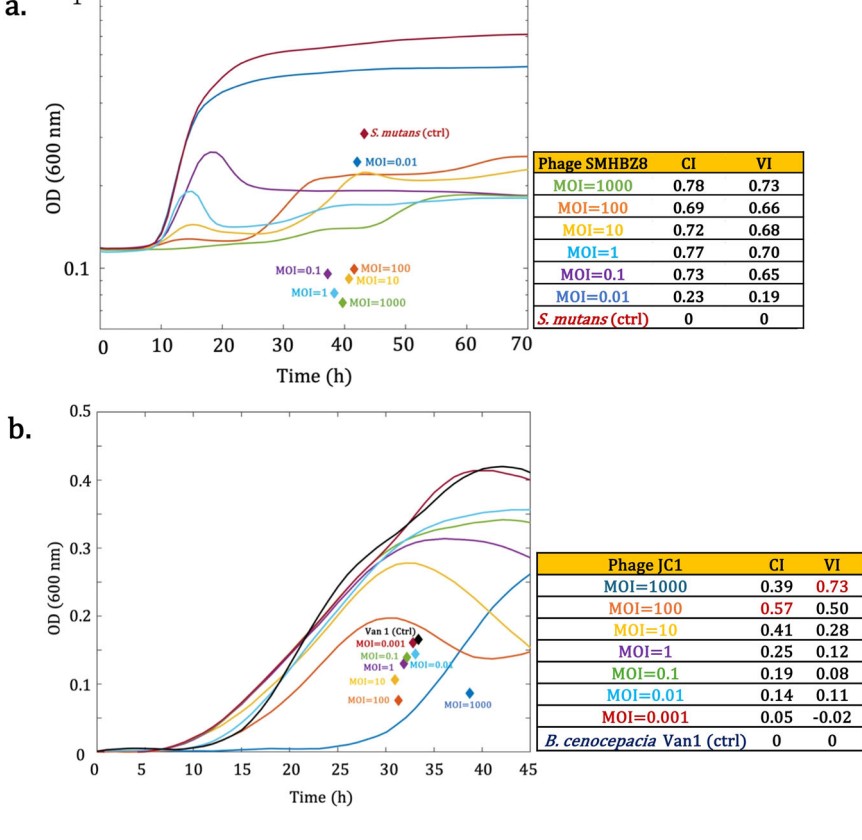

| Phage SMHBZ8 | CI | VI |
|---|---|---|
| MOI=1000 | 0.78 | 0.73 |
| MOI=100 | 0.69 | 0.66 |
| MOI=10 | 0.72 | 0.68 |
| MOI=1 | 0.77 | 0.70 |
| MOI=0.1 | 0.73 | 0.65 |
| MOI=0.01 | 0.23 | 0.19 |
| *S. mutans* (ctrl) | 0 | 0 |

| Phage JC1 | CI | VI |
|---|---|---|
| MOI=1000 | 0.39 | 0.73 |
| MOI=100 | 0.57 | 0.50 |
| MOI=10 | 0.41 | 0.28 |
| MOI=1 | 0.25 | 0.12 |
| MOI=0.1 | 0.19 | 0.08 |
| MOI=0.01 | 0.14 | 0.11 |
| MOI=0.001 | 0.05 | -0.02 |
| *B. cenocepacia* Van1 (ctrl) | 0 | 0 |

**Table 1 | The combination of the phage cocktails used & mentioned in this study**

| Cocktail | Phages and cofactors |
|---|---|
| Cocktail 1 | 44RR2.8t.2, 65.2, MQM1, Riv-10, SW69-9 |
| Cocktail 2 | 44RR2.8t.2, 65.2, MQM1, Riv-10, SW69-9 with 10 mM CaCl$_2$ |
| Cocktail 3 | 44RR2.8t.2, 65.2, MQM1, Riv-10, SW69-9.BK93 |
| Cocktail 4 | 44RR2.8t.2, 65.2, MQM1, Riv-10, SW69-9.BK93 with 10 mM CaCl$_2$ |
| Cocktail 5 | 44RR2.8t.2, 65.2, MQM1, Riv-10 |
| Cocktail 6 | 44RR2.8t.2, 65.2, MQM1, Riv-10 with 10 mM CaCl$_2$ |
| Four-phage cocktail[17] | 44RR2.8t.2, 65.2, Riv-10, SW69-9 |

*salmonicida* strains (Bacterium I and II, Fig. 6) were used for VI and CI calculations.

## Reporting summary

Further information on research design is available in the Nature Portfolio Reporting Summary linked to this article.

## Data availability

The raw data and details of calculations (equations) for VI and CI that support the findings of this study are available in supplementary data 1 file. The determination of the VI values were made using the procedures described in ref. 16. Raw data obtained by WebPlotDigitizer from growth curves belonging to refs. 28,29 are available from the corresponding author upon reasonable request. The WebPlotDigitizer used in this study, to extract OD and time values, is available at https://github.com/ankitrohatgi/WebPlotDigitizer. This tool allows users to extract the values of the coordinates from images of plots, charts, or graphs.

## Code availability

The Centroid_Index_Calculator software files for Windows and MacOS are deposited into Zenodo and are available at https://doi.org/10.5281/zenodo.11137800. The detailed guidelines for software installation are also provided in the same link, in addition to Supplementary Note 1.

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

## Acknowledgements

This research was funded by the Ministère de l'agriculture, des pêcheries et de l'alimentation du Québec (INNOVAMER Program), the Natural Sciences and Engineering Research Council of Canada (NSERC, grant number RGPIN-2019-04444), and Ressources Aquatiques Québec (RAQ). SM holds the Canada Research Chair in Bacteriophages. The authors gratefully acknowledge Valérie E. Paquet for providing phage SW69-9.BK93 as well as Moïra Dion and Geneviève Rousseau for testing the Centroid Index Calculator software and providing suggestions and feedback.

## Author contributions

N.H. performed experiments. N.H. and M.C. analyzed the data. M.C. and N.H. developed virulence metrics. M.C. developed the Centroid_Index_Calculator software. N.H. provided the Excel file calculations presented in Supplementary Data 1. N.H. and M.C. wrote the original draft. All the authors reviewed and edited the article. S.J.C. and S.M. supervised the project. S.J.C. and S.M. acquired funding. All authors have read and agreed to the published version of the manuscript.

## Competing interests

The authors declare no competing interests.
