## [Peer Review File · Communications Biology]

Reviewers' comments:

Reviewer #1 (Remarks to the Author):

Dear authors,

I read the manuscript carefully and found it was written in good language and organized well. However, I found no scientific novelty in the presented research. The only new proposal here is how the area under the curve (AUC) can be assessed better in some circumstances.

As you mentioned, there are previous works of VI or PhageScore (PS) to evaluate phage efficacy. In my work, I am mainly using PS so I will focus on this method described previously (Konopacki, M., Grygorcewicz, B., Dolegowska, B., Kordas, M. & Rakoczy, R. PhageScore: A simple method for comparative evaluation of bacteriophages lytic activity. *Biochem. Eng. J.* 161, 107652 (2020). <https://doi.org/10.1016/j.bej.2020.107652>).

The whole concept is based on the differences between the area under growth curves of the control and phage-infected populations. It was described how to use this to evaluate phage's lytic activity for a single case (PS) and between various MOI (PTS). In this manuscript, an analytical approach was proposed to calculate such areas. In your manuscript, you still using the same/similar concept modifying only how the area value can be obtained.

You wrote that in some cases analytical calculations proposed previously could be hard to obtain or can create errors when the functions are fit inaccurately. From my experience, I can agree with that, but many depend on numerical procedures, especially initial conditions to fit the functions properly. Having a problem with this, I am looking for other ways to calculate the area. Your method could be one of them, but still, some of the other researchers can bring new limitations for this in the future, so I don't see the point. The best option (and most accurate) in my opinion is to use some graphic software that can calculate the area under curves on a graph very fast and simply.

Nevertheless, having the values of the AUC (besides how you obtain them) you go back and calculate indexes described previously so you can analyze phage's lytic activity.

To conclude, in my opinion, I found the presented manuscript lacks scientific novelty because it modified only the way the AUC can be calculated, which can be better in some circumstances than methods described in previous manuscripts, however, as I wrote above, this could be done easier and better. The whole approach and idea remain unchanged.

Reviewer #2 (Remarks to the Author):

General: First point is, I think that the manuscript is written in a very clean and logical form, and I especially like how centroid index gives better prediction in comparison to virulence index, even when non-canonical growth curves are in question which are very problematic to handle. Well done.

1. The first comment is about the material and method section. We can agree that in last years phage research area is attracting more and more researchers worldwide, and for successfully implementation of centroid index software (or any other) is to assure that growing condition of the control is well performed. For instance, to obtain growth curves more than 96 h in a microplate format is not easily

done just with LB medium, you need to concentrate it (Super broth), adjust the pH and so on. I would like that that section is more elaborated and with more precise details so people can reproduce the experiment.

2. Second thing is that many people from different fields would like to try your software. It would be nice to have some sort of guideline of what can go wrong. For instance, as you mentioned this can be interesting to people from water research, and many times water phage samples can contain toxic compound that might show inhibition in the absence of phages which will lead to false positive readings. What are solutions to these problems? If authors can think about the problematic outcomes and incorporate it in the guideline it would be very useful for successful implementation of centroid index in the phage research.

Reviewer #3 (Remarks to the Author):

The authors present an interesting new method for assessing the growth characteristics of bacteria in the presence of a phage. Indeed up to date, there are no precise enough metrics to assess the growth characteristics. The proposed method is definitely an improvement to our current possibilities for assessing phage bacterium dynamics.

The results and discussion section starts with a whole explanation of what the approach is on a hypothetical model. This has in my view little added value and can be very much simplified and shortened. Also, it would be more interesting to use real data rather than original data. The authors can enrich by using publicly available growth curve data. These data will also allow the authors to see different general forms of the growth curves, and include multiple variations.

Testing of cocktails is indeed of interest, however it would be interesting to define what aspects of the interaction are of importance in a single phage experiment to be included in a cocktail.

In the discussion, the usefulness of the new parameter for assessing phage bacterium interaction dynamics would be good. To what purpose can this parameter be used, what are the implications for the development of clinical phage cocktails. How can the single phage bacterium interaction lead to a good cocktail? what is the value of your parameter for such inclusion. Finally, as mentioned above, it would be good to analyze also publicly available data.

Reviewer #4 (Remarks to the Author):

Summary of Review

Overall, this manuscript has promise for publication. The idea of having a useful measure for comparing virulence levels between uninfected controls versus viral infected cultures and/or virulence between one virus versus another virus on a given host has been an elusive target. The concept of using an index of virulence based on the centroid of a growth curve is intriguing. However, the manuscript as written does not provide the level of detail required to forward a convincing argument for the use of the proposed centroid based measure (CI).

FIRST, the manuscript would be strengthened by a detailed explanation of the biological significance of the centroid (or change/difference in centroid) within and between growth curves in terms of cell density/population within the culture.

SECOND, the manuscript would be strengthened by a clear explanation of the limitations of using CI to determine virulence in multi-phage/single-host (i.e., phage cocktail) infection assays. There is a bit of over-reach on the proposed utility of CI in host-competition assays.

THIRD, the manuscript would be strengthened by DIRECT COMPARISONS between CI measures versus specific growth (i.e., μ max) versus ISC versus AUC measures on a set of sample growth curves. The authors seem to have selected growth patterns which accommodate the centroid based measure rather than an objective comparison. This should be corrected and a side-by-side comparison should be provided for at least one set of experimental results.

FOURTH, the authors MUST provide data that convincingly shows that the resurgence in growth after administration of phage is indeed due to the emergence of bacteriophage resistance. The same types of growth curves with an initial peak and drop in cell culture followed by a steady logistic-like growth can also emerge by: (a) very low MOI in which the infection wanes without the emergence of truly resistant host genotypes; or, (b) by the presence of a non-lytic infection (i.e., blobbing/budding) whereby the initial input of viral suspension results in a dip in the culture until the virus establishes infection and then begins blebbling off virus. Taking culture from the far end of the infection time spectrum and re-culturing with high MOI phage to show that these hosts are truly phage-resistant (i.e., BIM as described in the manuscript) is essential.

One of the strengths of the proposal is the suggestion that a centroid-based measure can serve as a measure of virulence for cultures that consist of an evolving mixture susceptible and phage-resistant hosts (BIM). But this argument cannot be successfully made without addressing the concerns stated above.

Also, it is important that the authors use terminology that is common in the field and not create new jargon. The field is already overwhelmed with vague terminology. Very concise terminology with regard to concepts like: (a) lytic infection/replication versus non-lytic infection/replication; (b) viral resistant strain (and mechanism); (c) spot-on-lawn versus a serial dilution plaque assay or "halo" assay; and, so forth. Recent published works in the fields of bacteriophage biology, archaea virus biology, and eukaryotic (plant and animal) virus biology have begun an attempt to standardize terminology for virology and virus ecology. Conforming to these emerging norms would strengthen the manuscript.

Detailed Review

Abstract

Line 27 If there is room in the word count, please provide a sentence or clause on what limitations are overcome using CI and what does the centroid physically demonstrates.

Introduction

Line 38-39 end sentence and begin new sentence: ...found in the biosphere. Phage characterization has contributed

Line 42applications, which is often referred to as “phage therapy” (REFERENCE NEEDED)

Line 42 replace “They” with Bacteriophages are being studied worldwide.....
(note that it is not “now” - phages have been studied for decades - omit “now”)

Line 43 replace veterinary with “..... veterinary sciences.^{7,8}”

Line 44 change to: “The efficacy of a specific phage genotype to kill a host is often tested first by a spot.....”

Line 45 Omit: “The”. “...medium.⁹ Optical density (OD) measurements.....”

Line 46 change to: “...in the presence versus absence (uninfected control) of phage are also widely used.¹⁰”

Line 47 delete: “microplate” (it does not have to be a microplate reader.
The same thing can be done with a nanodrop or cuvette absorbance spectrometer. Also, delete “device”
– redundant

Line 48 change to: “....comparing the growth of uninfected bacteria against phage-treated bacteria under defined conditions of temperature and incubation time.

Line 49 change to: “In phage-bacteria infection studies, it is common for bacteriophage insensitive mutants (BIM) to emerge in an infected culture.”
(Plural acronyms such as BIMs should simply be BIM)

Line 52 change to: “...obstacles in phage therapy. It is also a factor that.....”

Line 54 change to: “...to assess phage virulence....”

Line 54 change: “standard” to “common” or “universally adopted”

Line 55-57 awkward: Phage virulence quantification facilitates comparing the lytic efficiency of different phages or phage cocktails against specific bacteria by providing an index value as

compared to visual qualitative approaches, which can be inconclusive 15,16.

(Rework sentence --- maybe make two sentences to express the thought?)

Line 58-59 awkward: Among available quantitative methods based on bacterial growth curve results, (long, run-on sentence....maybe break into several sentences for clarity)

Maybe change to:

Three quantitative methods used to assess phage efficiency from bacterial growth data are: area-under-the-curve; maximum specific growth rate μ_{max} ; and, curve fitting procedures.

Then proceed to explain each of these separate paragraphs.

Line 59 What is meant by “average behavior of a system”... this sentence should describe what area-under-the-curve (AUC) denotes biologically in terms of the number of host alive versus death or the number of cell divisions versus cell losses/deaths. Furthermore, the paper proposed to use the center of area, or centroid, which is the arithmetic mean position of a body – thus it also describes “average behavior of a system”

This is problematic throughout the paper. If the centroid is going to be used, then what is the biological significance of this measure in terms of cell population change.

Line 68-69 It is argued that the use of a centroid-based parameter may be the best option when a subsequent cycle of logistic growth occurs due to bacteriophage insensitive mutants arising in culture after a “crash”. This is where this argument should be clearly justified and explained.

Line 70 Should clarify what is meant by lysis.... Lysis is not simply “cell death”.. cells are dying throughout the duration of culture growth. Lysis is a rapid decline in net culture growth due to gross cell lysis. Lysis often requires the expression on lytic cycle proteins.

Line 74 What is meant by “...growth levels...” ? Does this refer to Nasymptote?

When a study is proposing to implement a new measure, it is important to be very specific about what is actually being considered as part of the measurement. Dissecting terminology is important and relating it to specific factors within an equation is also imperative.

Lines 83-97 This is where the argument for using centroid needs to be made very clear. What is the biological relevance in terms of host vs. host-infected virus populations? What is the proposed advantage of using a centroid-based measure? Why and how is this applicable to phage cocktails?

Results & Discussion

Line 123 Some error in Reference source shown

Line 130-131

In general, the idea of having a virulence indicator that accounts for the emergence of virus- or phage-resistant hosts during longer cycle/duration infections is meritorious. The utility of testing a virus or phage cocktail by analyzing host growth is limited. Although it provides some information regarding polymicrobial infection, unless the relative contribution of each pathogen on host growth is discernible, then it offers little in terms of biological significance. If the manuscript suggests that doing a sequential addition of phage to an initial single-phage/single-host infection assay, followed by a two-phage/single-host assay, then a three-phage/single-host infection assay will yield relative contributions of each phage to virulence, then this needs to be demonstrated.

Also in Figures 3, 4, and 7, what evidence is there that culture growth past timepoints (Fig. 3 – unknown; Fig. 4 – 40 hrs, Fig. 7 – 40 hrs) is actually bacteriophage resistant host. The same types of growth curves can be obtained by performing low MOI infection assays. Culture must be re-seeded with phage and show no detriment with respect to uninfected controls in order to demonstrate that these are really BIM. Showing changes in CRISPR sequences or the acquisition of a resistance gene (i.e., on a plasmid or within the host genome) may also serve as added verification of bacteriophage resistance. In Figure 6, the authors could strengthen the manuscript by showing how the putative data set outcomes and interpretation differ when ISC, μ_{max} , or phage score versus CI is used.

The centroid index (CI) described may have some utility; however, it needs to be clearly coupled to biology. Specifically, at any time point along the host growth curve, a measure of how many live cells is (i.e., cell density) determined using absorbance spectroscopy (i.e., optical density) as a proxy. A change from one time point to another show the net change in the number of new cells generated (e.g., via cell division) minus the number of cells lost (i.e., cell death) that occurred during t_2-t_1 . So, a concise understanding of slope-based measures between two time points (e.g., μ_{max}) and area-under-the-curve (AUC) is provided in a physiological context. It is also easy to determine detriment or facilitation of host growth using these measures since a difference can be calculated. The authors should expound upon what the centroid represents biologically. This is not clear in the manuscript. Furthermore, changes in the centroid from a control trial (e.g., uninfected host) versus an experimental trial (e.g., phage-infected) are clear if the difference occurs along a single axis (e.g., y-axis shift). The biological significance of the case where there is both an x-axis and y-axis shift is not clearly articulated.

It also seems that the authors have automatically assumed that what they are seeing in the psychrophilic phage system is lytic. The growth curves that presented are either non-lytic (i.e., blebbing/budding) or very low MOI traces where deep declines in culture are not occurring. The manuscript would benefit from showing a spot-on-lawn or “plaque” assay as well as any protein or mRNA based evidence of a lytic replication cycle.

Overall, the information presented in the manuscript is promising; however, the write-up needs clarification in several areas and a more detailed description of biological relevance of the centroid and changes in centroid. It is recommended that careful language regarding the utility of CI to multi-phage/single host interactions and interpretation be included in the manuscript.

DETAILED ANSWERS TO REVIEWERS (COMMSBIO-23-3183)

Our answers are in bold and in blue below the comments from the reviewers. The new text in the manuscript is between quotation marks and in *italic*. The line numbers (L. X) refer to the latest version of the manuscript.

Reviewer #1

1. I read the manuscript carefully and found it was written in good language and organized well. However, I found no scientific novelty in the presented research. The only new proposal here is how the area under the curve (AUC) can be assessed better in some circumstances. As you mentioned, there are previous works of VI or PhageScore (PS) to evaluate phage efficacy. In my work, I am mainly using PS so I will focus on this method described previously (Konopacki, M., Grygorcewicz, B., Dolegowska, B., Kordas, M. & Rakoczy, R. PhageScore: A simple method for comparative evaluation of bacteriophages lytic activity. *Biochem. Eng. J.* 161, 107652 (2020). <https://doi.org:10.1016/j.bej.2020.107652>). The whole concept is based on the differences between the area under growth curves of the control and phage-infected populations. It was described how to use this to evaluate phage's lytic activity for a single case (PS) and between various MOI (PTS). In this manuscript, an analytical approach was proposed to calculate such areas. In your manuscript, you still using the same/similar concept modifying only how the area value can be obtained.

Many thanks to the referee for this comment, which allowed us to see that there could be potential confusion in parts of the text for some readers. While two regions may have identical areas under the curve, as illustrated in Figure 6 of the manuscript, differences in the distribution of the OD data within each region will lead to different centroids or centers of gravity. Consequently, the concepts of centroid (or center of gravity) and the area under the curve (AUC) are inherently distinct. The centroid index provides a higher level of precision due to its capability to incorporate OD curve trend in its calculation, contrasting with the AUC method which is solely based on the average value of OD curve within the time interval of the experiment. Therefore, our focus lies in characterizing the dynamics of variation of OD over time, emphasizing the distribution of these variations rather than solely considering average values, as is typically done with the AUC method.

In detail, the centroid or center of gravity of a region, which forms the basis for the centroid index (CI), is mathematically defined using the concept of the "first moment of inertia", as outlined in the manuscript, below the Eq. (1): "*The first moment of area is associated with the topological distribution of an area with respect to an axis*". In other words, centroid index is constructed based on the topological distribution of the OD points. This parameter is a well-known concept in mathematics but, to our knowledge, not explored in biological contexts.

2. You wrote that in some cases analytical calculations proposed previously could be hard to obtain or can create errors when the functions are fit inaccurately. From my experience, I can agree with that, but many depend on numerical procedures, especially initial conditions to fit the functions properly. Having a problem with this, I am looking for other ways to calculate the area. Your method could be one of them, but still, some of the other researchers can bring new limitations for this in the future, so I don't see the point. The best option (and most accurate) in my opinion is to

use some graphic software that can calculate the area under curves on a graph very fast and simply. Nevertheless, having the values of the AUC (besides how you obtain them) you go back and calculate indexes described previously so you can analyze phage's lytic activity.

To conclude, in my opinion, I found the presented manuscript lacks scientific novelty because it modified only the way the AUC can be calculated, which can be better in some circumstances than methods described in previous manuscripts, however, as I wrote above, this could be done easier and better. The whole approach and idea remain unchanged.

Thanks to the referee for raising this point about problems encountered when dealing with calculating phage score, and how sometimes AUC calculations fail. We have highlighted the main issues with AUC in our response to the reviewer's comment #1 above and also throughout the manuscript (Figures 6, 7, 9, 10, and 12). Here, we discuss the phage score method.

The phage score method (doi.org/10.1016/j.bej.2020.107652) assumes that bacterial growth can be represented via the “logistic growth” model, that is:

$$f_1(t) = y_0 + \frac{a}{1 + e^{-\frac{t-x_0}{d}}}, \text{ Eq.(A)}$$

and the phage-treated bacteria can be modelled using a “generalized double exponential logistic” equation, that is:

$$f_2(t) = \frac{L}{1 + b e^{-kt} + c e^{ht}}. \text{ Eq.(B)}$$

First of all, many growth curves do not follow these patterns that mathematical modelling dictates, they may or may not behave as the logistic model interpolates. Therefore, these fundamental assumptions are simply not true for noncanonical curves, as shown in Figure 4 in the manuscript. Secondly, the phage score method requires the curve fitting procedure in order to obtain the unknowns which are y_0 , a , x_0 in Eq. (A) and L , b , k , c , h in Eq. (B).

Nevertheless, the curve fitting procedure, intrinsically and inevitably, induces numerical errors. Moreover, the unknowns in Eqs. (A, B) obtained via curve fitting are not unique or absolute, and “*This lack of uniqueness is by itself a reason to question any mechanistic interpretation of growth parameters obtained by curve fitting alone*”, as stated by others (see doi.org/10.1080/10408398.2011.570463). Hence, although the phage score method prevents numerical errors associated with trapezoidal integration used in AUC, it introduces other errors (associated with the growth model assumption and curve fitting). These errors may outweigh the error resulted from trapezoidal integration. Here in Figure R1, we provide an example in which we have calculated phage score using the raw data of Figure 7a . The obtained values are presented in Figure R2.

Figure R1. Curve fitting applied on bacterial growth curves related to Figure 7a in the manuscript. **A.** Curve fitting applied on the curve of bacteria only. **B.** Curve fitting applied on the curve when a four-phage cocktail was added at a MOI of 0.1. **C.** Curve fitting applied on the curve when a four-phage cocktail was added at a MOI of 1. **D.** Curve fitting applied on the curve when a four-phage cocktail was added at a MOI of 10. Fitting process was conducted according to the “logistic growth” equation for the control bacteria, and the “generalized double exponential logistic” equation for the phage-treated bacteria curves, followed by the analytical integration. All curve fitting process were performed using MATLAB.

Methods	Virulence Index	Phage Score	Centroid Index
MOI=0.1	0.56	0.56	0.55
MOI=1	0.53	0.52	0.35
MOI=10	0.62	0.59	0.23
Ctrl bacteria	0	0	0

Figure R2. Comparison between values calculated by three methods, Virulence Index, PhageScore, and Centroid Index. The values calculated for each method in presented in the table. Values obtained by Virulence Index and PhageScore methods do not account for the regrowth of bacteria as a result of incubation with the highest MOI (MOI of 10), since both of these methods showed the highest scores. Such high scores would (mistakenly) indicate the most efficient cocktail, compared to other MOIs (0.1 & 1) (solid yellow rectangle). However, Centroid Index could distinguish the regrowth of bacteria, giving the lowest value to MOI=10 compared with other conditions (hollow red rectangle).

This example reinforces the issue that the curves do not follow the logistic model. Also, similar to AUC, the phage score fails to assess the lytic efficacy of the examples in Figure7a.

The referee mentions that “... *I am looking for other ways to calculate the area. Your method could be one of them, but still, some of the other researchers can bring new limitations for this in the future, so I don't see the point.*” While we agree that others may come up with alternative options later, our proposed application of the centroid method is still novel and very efficient in a phage context. Also, the software provided in this manuscript (Supplementary material 1, Figure S1, and Supplementary material 1) facilitates the centroid index calculations for researchers in the field. It is possible to obtain the CI values for many conditions in only few minutes. We therefore consider that the proposed method resolves issues not considered by other calculation methods as well as provides, a reliable and rapid quantitative analysis of phage activity.

Reviewer #2:

1. General: First point is, I think that the manuscript is written in a very clean and logical form, and I especially like how centroid index gives better prediction in comparison to virulence index, even when non-canonical growth curves are in question which are very problematic to handle. Well done.

We are happy to see that the reviewer appreciates our work to propose a new method to quantify the effectiveness of phages.

2. The first comment is about the material and method section. We can agree that in last year's phage research area is attracting more and more researchers worldwide, and for successfully implementation of centroid index software (or any other) is to assure that growing condition of the control is well performed. For instance, to obtain growth curves more than 96 h in a microplate format is not easily done just with LB medium, you need to concentrate it (Super broth), adjust the pH and so on. I would like that that section is more elaborated and with more precise details so people can reproduce the experiment.

We totally agree that it is key to have a medium in which the bacterium of interest grows well under the experimental conditions. We can understand that this was perhaps not clear enough in the first version of the manuscript. Of note, *A. salmonicida* subsp. *salmonicida* is a psychrophilic bacterium that cannot grow at temperatures normally exceeding 20 °C. Its growth is slow and there is no issue of evaporation because of the low temperatures. Regular growth media for this bacterium are LB, TSB, or HL5, in which *A. salmonicida* grows for at least 72h (doi.org/10.1093/femsle/fnx239, and doi.org/10.1186/s12864-016-2381-3). Therefore, it is also not necessary to use richer media. If we had used *E. coli* for example, we would have needed shorter period time (24 or 48 h). We added this information in the manuscript:

*“In a previous study, we tested a four-phage cocktail at different Multiplication of Infection (MOI) on two different strains of *A. salmonicida* subsp. *salmonicida* ([doi.org:10.3390/v13112241](https://doi.org/10.3390/v13112241)). This bacterial species is psychrophilic with optimal growth around 20 °C. Given its relatively slow growth, even slower than other mesophilic *A. salmonicida* at this temperature ([doi: 10.1093/femsle/fnx239](https://doi.org/10.1093/femsle/fnx239), and doi.org/10.1186/s12864-016-2381-3), growth experiments were*

carried out over longer periods up to 96 hours (doi.org:10.3390/v13112241). We noticed that the use of VI to estimate phage virulence with our data was problematic.” (Lines 261-266).

3. Second thing is that many people from different fields would like to try your software. It would be nice to have some sort of guideline of what can go wrong. For instance, as you mentioned this can be interesting to people from water research, and many times water phage samples can contain toxic compounds that might show inhibition in the absence of phages which will lead to false positive readings. What are solutions to these problems? If authors can think about the problematic outcomes and incorporate it in the guideline it would be very useful for successful implementation of centroid index in the phage research.

It is necessary to point out that these experiments are not for raw water sample. Even though *A. salmonicida* is a problematic bacterium in aquaculture, the experiments were performed here using pure cultures of both bacteria and phage. Water samples were, initially obtained from environmental sources to isolate phages or bacteria. After performing isolation and purification processes to obtain those phage/bacteria isolates, the phage lytic efficiency on the desired bacteria culture was evaluated (doi.org/10.1093/femsle/fnv002, doi.org/10.1016/j.virusres.2023.199165, doi.org/10.1038/s41598-017-07401-7, and doi.org/10.3390/v13112241). Therefore, here we are not dealing with environmental samples bearing variety of impurities, probable toxic compounds, etc., as mentioned by the reviewer. The focus of this investigation is solely on examining the efficiency of phages or phage cocktails on the bacterium of interest. It is an attempt to quantify such interaction via CI. The phage infection generally depends on variety of factors, including the phage type, bacterial strain, culture type/condition, MOI, presence of divalent cations like calcium (Ca²⁺) or magnesium (Mg²⁺), experimental design or type of assay, duration of the experiment, and other factors. For this part, we agree on the fact that a one-size-fits-all experimental guideline may not be applicable for all types of phage-bacteria interaction. It is after the optimization of a desired experimental condition and obtaining the growth curves that indices like VI or CI are applicable for comparative purposes. These tools can therefore help researchers in variety of ways including phage screening, evaluation of phage strains, detection of mutants, infection conditions and/or the susceptibility of host strains, and the formulation of phage cocktails (doi.org/10.1089/phage.2019.0001).

Reviewer #3

1. The authors present an interesting new method for assessing the growth characteristics of bacteria in the presence of a phage. Indeed, up to date, there are no precise enough metrics to assess the growth characteristics. The proposed method is definitely an improvement to our current possibilities for assessing phage bacterium dynamics.

Thank you very much for your comment . Indeed, there are currently no indices that can be simultaneously and universally applied to all types of growth curves, meaning canonical or non-canonical curves, and this why CI is proposed.

2. The results and discussion section starts with a whole explanation of what the approach is on a hypothetical model. This has in my view little added value and can be very much simplified and shortened. Also, it would be more interesting to use real data rather than original data. The authors can enrich by using publicly available growth curve data. These data will also allow the authors to see different general forms of the growth curves and include multiple variations.

We appreciate your suggestion to use publicly available data. Using a WebPlotDigitizer <https://apps.automeris.io/wpd/>, we extracted the OD and time data of the growth curves from two recent studies, and recalculated the VI values, in addition to evaluating the CI values. Our VI values corresponded with the reference values. After confirmation of VI values, using MATLAB, we illustrated the curves, showing the location of centroids of each curve with colored diamonds. These figures are now added to the manuscript. Please check the Results and Discussion section, lines 340-387, and Figures 11 and 12. Regarding the text, since Reviewer 4, requested more info, we didn't shorten this section.

Testing of cocktails is indeed of interest; however, it would be interesting to define what aspects of the interaction are of importance in a single phage experiment to be included in a cocktail. In the discussion, the usefulness of the new parameter for assessing phage bacterium interaction dynamics would be good. To what purpose can this parameter be used, what are the implications for the development of clinical phage cocktails. How can the single phage bacterium interaction lead to a good cocktail? what is the value of your parameter for such inclusion. Finally, as mentioned above, it would be good to analyze also publicly available data.

We can agree that in phage/bacteria interaction, the regrowth of bacterial subpopulation, or uprising OD trajectories of bacterial cultures infected by a phage/phage cocktail usually represents the decreased efficiency of that phage against the bacterial host (doi.org/10.1073/pnas.2104592118, doi.org/10.1128/spectrum.02072-22).

One significant point emphasized by the CI method is the attribution of a lower score (representing lower lytic efficiency) to a phage or a phage cocktail when their interaction with the bacteria of interest leads to bacterial regrowth. So, one of the main utilizations of CI method could be to screen *in vitro* the efficacy of a given phage or phage cocktail to kill the targeted bacterial strain. In case of therapeutic or biocontrol purposes, the regrowth of bacterial population after extended incubation is not a desired outcome. Before this study, we felt that there was no quantitative index in the literature that could precisely consider the bacterial regrowth towards the end of phage-bacteria incubation time. The publicly available data from doi.org/10.3390/v14050938 and doi.org/10.3390/v13050825 are now added to the manuscript, which you can find in Figures 11 and 12.

Reviewer #4

Overall, this manuscript has promise for publication. The idea of having a useful measure for comparing virulence levels between uninfected controls versus viral infected cultures and/or virulence between one virus versus another virus on a given host has been an elusive target. The concept of using an index of virulence based on the centroid of a growth curve is intriguing. However, the manuscript as written does not provide the level of detail required to forward a convincing argument for the use of the proposed centroid based measure (CI).

1. The manuscript would be strengthened by a detailed explanation of the biological significance of the centroid (or change/difference in centroid) within and between growth curves in terms of cell density/population within the culture.

Thanks to the referee for bringing up the matter of biological significance of the CI. This point has been added to the manuscript.

“Biological significance of CI

We can conceptualize a bacterial growth curve in geometric terms for analytical purposes since it shows geometrical properties. In a bacterial growth curve, we typically plot time on the x-axis and the optical density, representing the population density, on the y-axis. This creates a curve that represents the change in bacterial population over time. The bacterial growth curve itself has a shape that changes over time, starting with a lag phase, followed by an exponential growth, then reaching a plateau during the stationary phase, and potentially declining during the death phase. We can think of the centroid as a hypothetical point that represents the "center of mass" or "balance point" of the growth curve. It is not a physical point within the curve, but a conceptual one, that summarizes the distribution of the bacterial population densities over time. In other words, the position of the centroid can indicate where the growth is centered on the bacterial growth curve, analogous to how it represents the balance point of a geometric object (center of gravity of a physical object). Changes in the position of the centroid indicate shifts in the distribution of cell population densities over time. This particularly facilitates the comparison between the bacterial growth affected by a unique phage or a combination of phages, since each growth curve may exhibit unique trends due to a number of reasons (type of phages, mutations in bacterial host, etc.). The centroid of any growth curve, having two coordinates (\bar{x} and \bar{y}) provides information regarding:

- 1. The magnitude of cell density or growth (\bar{y})*
- 2. The timing of when this density occurs (\bar{x} or \bar{t}).*

For example, \bar{t} shifting towards later times might indicate the regrowth of the bacterial population, and the upward \bar{y} might represent a higher magnitude of cell density or the bacterial regrowth. So far, there are no other methodologies which pinpoint where the cell density/growth is located, and the regrowth of the bacterial population is mostly not adequately or not at all considered by current indices (see Figure 7a). Overall, by considering the locations and timings of the peaks, the centroid index (CI) provides additional information regarding the phage-bacteria dynamics over time.”

(Lines 233-258).

2. The manuscript would be strengthened by a clear explanation of the limitations of using CI to determine virulence in multi-phage/single-host (i.e., phage cocktail) infection assays. There is a bit of over-reach on the proposed utility of CI in host-competition assays.

Thank you for your comment. This section is now added to the manuscript:

“In summary, the existing virulence indices provide a means to standardize and quantify the interaction between phages and bacteria. These comparative tools can provide guidance in selection of potential phages in specific applications (phage therapy or biocontrol applications) or aiding in the screening and optimization of phage candidates ([doi.org:10.1016/j.bej.2020.107652](https://doi.org/10.1016/j.bej.2020.107652), [doi.org:10.1089/phage.2019.0001](https://doi.org/10.1089/phage.2019.0001)). However, virulence in

multi-phage/single-host systems can involve highly dynamic fluctuations over time, influenced by a variety of factors such as phage-host coevolution, mutational resistance, counter-resistance ([doi.org:10.1007/978-3-319-41986-2_7](https://doi.org/10.1007/978-3-319-41986-2_7), [doi.org:10.1111/evo.13731](https://doi.org/10.1111/evo.13731)), competition between phage variants during incubation time, etc. ([doi.org:10.1038/s41564-022-01157-1](https://doi.org/10.1038/s41564-022-01157-1)). The centroid index method considers these temporal fluctuations. The centroid index or any other available indices may not capture all the detailed fluctuations of a growth curve and therefore depending solely on these indices may overlook subtle nuances in phage behavior. Nevertheless, by presenting examples derived from our research as well as from existing literature, we have illustrated the robustness of the CI method in validating both canonical and non-canonical growth curves, underscoring the efficacy and reliability of this method, especially when a gradual increase in OD occurs after prolonged incubation. (Lines 389-402).

3. The manuscript would be strengthened by DIRECT COMPARISONS between CI measures versus specific growth (i.e., μ_{max}) versus ISC versus AUC measures on a set of sample growth curves. The authors seem to have selected growth patterns which accommodate the centroid based measure rather than an objective comparison. This should be corrected, and a side-by-side comparison should be provided for at least one set of experimental results.

In order to calculate I_{sc} value, there should be $N_{asymptote}$ (the peak host growth) for a particular growth curve ([doi.org:10.1099/jgv.0.001515](https://doi.org/10.1099/jgv.0.001515)). This is why we already discussed this issue in the manuscript.

*“Methods such as I_{sc} ([doi.org:10.1099/jgv.0.001515](https://doi.org/10.1099/jgv.0.001515)) are based on the assumption that bacterial growth curves would eventually approach a stationary phase or a maximum growth level. In fact, the biological data may not always respect this principle since the presence of phages may have an impact only for a certain period of time or have a partial effect without producing a canonical curve. Unless a distinct stationary phase or maximum growth level has been obtained during the experiment, the calculated asymptotic number or ratio would be an extrapolated value that is not representative of the biological situation. It is known that the dynamics of a complex system, where many interactions (and strong nonlinearities) are involved, is not always predictable far into the future (Moon, F. C. 2008. *Chaotic and fractal dynamics: introduction for applied scientists and engineers.* (John Wiley & Sons).” (Lines 168-175).*

The graphical representation of the uncertainty of asymptotic growth ($N_{asymptote}$) prediction is illustrated in Figure 4 (lines 186-192). Additionally, the same point regarding $N_{asymptote}$ was pointed out elsewhere:

“Estimation of $N_{asymptote}$ (“ N_{max} ”) or corresponding growth ratio by regression can be tricky. If a clearly discernible “stationary phase” has not been reached during the experiment, the asymptotic number or ratio calculated in this way would be an extrapolated value.” ([doi.org:10.1080/10408398.2011.570463](https://doi.org/10.1080/10408398.2011.570463))

The fact that $N_{asymptote}$ is not applicable for non-canonical curves (curves not reaching to stationary phase) is already acknowledged:

“It is generally agreed that a reasonable upper bound is the beginning of stationary phase or peak growth (i.e. $N_{asymptote}$). However, non-canonical host growth during infection may render this value difficult to determine.” ([doi.org:10.1099/jgv.0.001515](https://doi.org/10.1099/jgv.0.001515))

Therefore, even though suitable for canonical growth curves, this particular methodology is not applicable for non-canonical types of curves, as observed with *Aeromonas salmonicida* growth curves (see Figure 7a in the manuscript (lines 285-293)).

The issue for μ max is also discussed in the manuscript in Figure 5, and in the manuscript (lines 205-214).

Therefore, the rationale behind our exclusion of certain methods (I_{sc} , $N_{asymptote}$, μ max) for assessing growth curves stemmed from their lack of suitability or relevance to our experimental conditions.

Regarding PhageScore and AUC (Virulence Index) methods and the comparison of their results with CI, We already assessed the growth curves of phage-infected *A. salmonicida* strains with these methods, and we could not reach to a conclusive result, since the estimations by PhageScore were really close to the AUC method (for PhageScore, see Figures R1 and R2 at page 3 of this document). For AUC or VI method, see Figures 6, 7, 8, 9, 10 (our experimental data) as well as Figures 11 and 12 (data obtained from literature) in the manuscript.

It is important to mention that it is for the cases of non-canonical curves that values of PhageScore or VI do not correspond to the true biological meaning of these curves. This is a significant discrepancy which we tried to address in this manuscript. For the canonical curves, the values obtained by CI is very close to what is obtained with PhageScore or VI.

4. The authors MUST provide data that convincingly shows that the resurgence in growth after administration of phage is indeed due to the emergence of bacteriophage resistance. The same types of growth curves with an initial peak and drop in cell culture followed by a steady logistic-like growth can also emerge by: (a) very low MOI in which the infection wanes without the emergence of truly resistant host genotypes; or, (b) by the presence of a non-lytic infection (i.e., blobbing/budding) whereby the initial input of viral suspension results in a dip in the culture until the virus establishes infection and then begins blobbing off virus. Taking culture from the far end of the infection time spectrum and re-culturing with high MOI phage to show that these hosts are truly phage-resistant (i.e., BIM as described in the manuscript) is essential.

Indeed, very low MOI, non-lytic infection, mutants, etc., can compromised phage therapy. While not shown here, we previously generated *Aeromonas salmonicida* subsp. *salmonicida* BIM, using myophage SW69-9 ([doi:10.1111/mmi.14308](https://doi.org/10.1111/mmi.14308)). The genome analysis on the isolated BIM led to identify mutation in genes involved in the biogenesis of lipopolysaccharides (LPS) and on an uncharacterized gene (*ASA_1998*). However, it must be considered that the purpose of the suggested method in this manuscript is to improve the current available methods for the evaluation of bacterial growth curves in presence of phages, especially when curves are mostly non-canonical. Please also check the answer to the next question below.

5. One of the strengths of the proposal is the suggestion that a centroid-based measure can serve as a measure of virulence for cultures that consist of an evolving mixture susceptible and phage-resistant hosts (BIM). But this argument cannot be successfully made without addressing the concerns stated above.

Thank you again for mentioning this point. After reading reviewers' comments, we realized that we wrongly placed too much emphasis on BIMs in the first version of the manuscript. So, we replace BIM mentions in the manuscript simply by "bacterial regrowth" to avoid over interpretation of the curves. We agree with the reviewer that many reasons other than just BIMs can explain these OD increases. Regardless of why bacteria are regrowing (or why the OD is increasing) the fact remains that CI works in any situation (canonical or non-canonical), and the objective of this study is the introduction of the centroid index method.

6. Also, it is important that the authors use terminology that is common in the field and not create new jargon. The field is already overwhelmed with vague terminology. Very concise terminology with regard to concepts like: (a) lytic infection/replication versus non-lytic infection/replication; (b) viral resistant strain (and mechanism); (c) spot-on-lawn versus a serial dilution plaque assay or "halo" assay; and, so forth. Recent published works in the fields of bacteriophage biology, archaea virus biology, and eukaryotic (plant and animal) virus biology have begun an attempt to standardize terminology for virology and virus ecology. Conforming to these emerging norms would strengthen the manuscript.

Thank you for the comment. "BIM" has now been changed to "bacterial regrowth" throughout the manuscript. In lines 42-43, we also changed "spot test assay" to "spot-on-lawn assay".

Detailed Review

Abstract

7. Line 27 If there is room in the word count, please provide a sentence or clause on what limitations are overcome using CI and what does the centroid physically demonstrate.

The journal requests that the abstract not exceed 150 words. Our first version of the abstract was over 180 words. We have shortened the abstract and tried to cover the reviewer's request in the few words available.

Introduction

8. Line 38-39 end sentence and begin new sentence: ...found in the biosphere. Phage characterization has contributed ...

Modification is done as suggested (currently, lines 35-36).

9. Line 42applications, which is often referred to as "phage therapy" (REFERENCE NEEDED)

Modification is done as suggested. A reference (<https://doi.org:10.2147/BTT.S381237>) was added as reference number 4 (line 39).

10. Line 42 replace “They” with Bacteriophages are being studied worldwide.....
(note that it is not “now” - phages have been studied for decades - omit “now”)

Thank you for the relevant comment. Modified as suggested (line 39).

11. Line 43 replace veterinary with “..... veterinary sciences.7,8.”

Modified as suggested (line 41).

12. Line 44 change to: “The efficacy of a specific phage genotype to kill a host is often tested first by a spot.....”

We have decided to consider your suggestion only partially by adding the word "specific", but not the word "genotype", since we think this word is incomprehensible in this particular sentence (line 42).

13. Line 45 Omit: “The”. “...medium.9 Optical density (OD) measurements.....”

Modified as suggested (line 43).

14. Line 46 change to: “...in the presence versus absence (uninfected control) of phage are also widely used.10

It is now changed (line 44).

15. Line 47 delete: “microplate” (it does not have to be a microplate reader. The same thing can be done with a nanodrop or cuvette absorbance spectrometer. Also, delete “device” – redundant.

Thank you, however, even though these devices also capture the OD, to our knowledge, they are not well adapted to perform phage-bacteria incubation for a long period of time to yield a growth curve showing the OD versus time since this type of experiments requires shaking for oxygenation and to prevent biomass decantation.

16. Line 48 change to: “...comparing the growth of uninfected bacteria against phage-treated bacteria under defined conditions of temperature and incubation time.

Thank you for your suggestion. It was modified accordingly except that we kept “with” instead of “against” (lines 46-47).

17. Line 49 change to: “In phage-bacteria infection studies, it is common for bacteriophage insensitive mutants (BIM) to emerge in an infected culture.” (Plural acronyms such as BIMs should simply be BIM)

Following the feedback from the reviewers, we have come to recognize that the emphasis on the term “BIM” throughout the manuscript has inevitably led to deviations regarding the purpose of this work. Therefore, BIM is no longer present in the manuscript and replaced by “bacterial regrowth”.

18. Line 52 change to: “...obstacles in phage therapy. It is also a factor that.....”

Modified as suggested (lines 49-50).

19. Line 54 change to: “....to assess phage virulence....”

Modified as suggested (line 52).

20. Line 54 change: “standard” to “common” or “universally adopted”

Thanks for your suggestions. Changed to “common” (line 52).

21. Line 55-57 awkward: Phage virulence quantification facilitates comparing the lytic efficiency of different phages or phage cocktails against specific bacteria by providing an index value as compared to visual qualitative approaches, which can be inconclusive 15,16.
(Rework sentence --- maybe make two sentences to express the thought?)

Many thanks for pointing out this issue. Sure. It is now changed to:

“Quantifying phage virulence facilitates the comparison of the lytic efficiency of various phages or phage cocktails against specific bacteria. This is achieved by providing an index value, in contrast to visual qualitative approaches, which may be inconclusive” (lines 53-55).

22. Line 58-59 awkward: Among available quantitative methods based on bacterial growth curve results, (long, run-on sentence....maybe break into several sentences for clarity)
Maybe change to: Three quantitative methods used to assess phage efficiency from bacterial growth data are: area-under-the-curve; maximum specific growth rate μ_{max} ; and, curve fitting procedures. Then proceed to explain each of these separate paragraphs.

Thank you for the suggestion. See the modified version in the text:

“Among available quantitative methods for studying bacterial growth curves, the virulence index (VI) stands out by mainly focusing on the average behavior of the system through the calculation of the area under the curve” (lines 56-58).

23. Line 59 What is meant by “average behavior of a system”... this sentence should describe what area-under-the-curve (AUC) denotes biologically in terms of the number of host alive versus death or the number of cell divisions versus cell losses/deaths. Furthermore, the paper proposed to use the center of area, or centroid, which is the arithmetic mean position of a body – thus it also describes “average behavior of a system”

The average behavior of the system means that VI can evaluate the average magnitude of a bacterial population, without necessarily pinpointing where the higher levels of cell density are observed, and without considering the trend of the curve. CI is not equivalent with the average behaviour of the system, since emphasizes on the distribution of cell density during time, which AUC does not. The distribution of OD points is not considered in AUC, while it is in CI. In other words, CI focuses on capturing the dynamics of the cell population change over time, and any alterations in the system are reflected in the CI value. In contrast, AUC is incapable of considering such changes. The clear example of this incapacity is shown in Figure 6. In this figure, while two types of growth curves show considerably different trend during the same incubation time, their average behaviour of system or AUC resulted in the same value. Thus, the AUC can cause interpretation errors which are resolved by CI. Please also check the response to the next question below, and also the response to Reviewer #1.

24. This is problematic throughout the paper. If the centroid is going to be used, then what is the biological significance of this measure in terms of cell population change.

Please check the complete answer on page 5 of this document, the answer to the Question #1 of Reviewer #4.

25. Line 68-69 It is argued that the use of a centroid-based parameter may be the best option when a subsequent cycle of logistic growth occurs due to bacteriophage insensitive mutants arising in culture after a “crash”. This is where this argument should be clearly justified and explained.

The main objective of this study is to discuss a methodology capable of overcoming the problems previously found for other methods. The core idea, and why this methodology was developed from the first place was our own problem with current methods not being able to explain the type of non-canonical curves we had with a bacterium which takes more than usual time to grow. As explained previously, we are not trying to justify the type of mutations. In fact, as mentioned above, we recognize that it was our error, in the first version of the manuscript, to speak specifically of BIMs to explain the increase in OD towards the end of the incubation period. We have eliminated this aspect of this new version of the manuscript to focus solely on the mathematical description of the new proposed method.

While it is a very interesting topic to investigate further in an independent study in future, the primary purpose of suggesting the CI method is not necessarily to confirm whether the OD increase, especially towards the end of the bacterial growth curve, is a direct result of BIM population growth, or it is due to another phenomenon. In contrast, the objective of developing such a method is to equipped phage researchers with a tangible method, enabling them to have values which could correctively consider factors impeding the lysis effect expected to be seen in therapeutic purposes using phages. These results have emerged from our analysis of our own growth curves, data extracted from relevant literature, and feedback provided by colleagues who have employed our methodology. After observing how their phage of interest affected the host strain in liquid culture, and after attributing a proper quantitate value to it, it will be time to investigate what exact type of mutation in bacterial strain had led to the outburst and regrowth of bacterial population. This later objective is beyond the purpose of CI method because there could be many reasons for the increase in OD values during time.

26. Line 70 Should clarify what is meant by lysis.... Lysis is not simply “cell death”.. cells are dying throughout the duration of culture growth. Lysis is a rapid decline in net culture growth due to gross cell lysis. Lysis often requires the expression on lytic cycle proteins.

Your comment made us realize to change the text. Our study concerns a mathematical method which seeks to make the analyzes of trends in the curves more precise. We have therefore removed our attempt to explain the variation of curves in this part of the text which now reads as follows:

“An example of non-canonical growth curves, in which the steady state may not be approached, is when there is an increase in the OD linked to the bacterial regrowth. Other examples are variations in OD halfway through an experiment in the condition where phages were included as we observed in a previous study ([doi.org:10.3390/v13112241](https://doi.org/10.3390/v13112241)), or the OD decrease of the control conditions, particularly in experiments with a long incubation period ([doi.org:10.3390/v13112241](https://doi.org/10.3390/v13112241)).” (Lines 66-70).

27. Line 74 What is meant by “...growth levels...”? Does this refer to $N_{asymptote}$? When a study is proposing to implement a new measure, it is important to be very specific about what is actually being considered as part of the measurement. Dissecting terminology is important and relating it to specific factors within an equation is also imperative.

Thank you for your comment. Yes, the growth levels here refer to the “ $N_{asymptote}$ ”. In the context of bacterial growth curves, the “ $N_{asymptote}$ ” usually refers to the plateau phase of growth, where the population density reaches a maximum level and when the growth stabilizes. We added this word in parenthesis in the sentence mentioned in line 73.

28. Lines 83-97 This is where the argument for using centroid needs to be made very clear. What is the biological relevance in terms of host vs. host-infected virus populations? What is the proposed advantage of using a centroid-based measure? Why and how is this applicable to phage cocktails?

The biological significance of the centroid index, and the advantages of this method to other available methods have already been addressed when answering previous reviewers. Please check our answer to your first comment. For the last part of the comment, it is important to mention that CI or any virulence index could be applied to interpret the growth curve under the effect of any phage combination, whether it is individual phage or phage cocktail. For the examples of applicability of CI on different conditions, please check Figures 7, 8, 9, and 10 for the effect of variety of phage cocktails on different bacterial strains (*A. salmonicida* strains), and please check Figures 11 and 12 for the effect of individual phage on different bacterial species (*Streptococcus mutans* and *Burkholderia cenocepacia* Van1).

Results & Discussion

29. Line 123 Some error in Reference source shown.

Thank you very much for mentioning this point. The error is now fixed (line 122).

30. Line 130-131

In general, the idea of having a virulence indicator that accounts for the emergence of virus- or phage-resistant hosts during longer cycle/duration infections is meritorious. The utility of testing a virus or phage cocktail by analyzing host growth is limited. Although it provides some information regarding polymicrobial infection, unless the relative contribution of each pathogen on host growth is discernible, then it offers little in terms of biological significance. If the manuscript suggests that doing a sequential addition of phage to an initial single-phage/single-host infection assay, followed by a two-phage/single-host assay, then a three-phage/single-host infection assay will yield relative contributions of each phage to virulence, then this needs to be demonstrated.

Thank you for the comment. This question has been already answered above as we pointed out that the main objective of this study is a proposal of a new quantitative virulence index, with the emphasize on comparing it with other available methods.

31. Also in Figures 3, 4, and 7, what evidence is there that culture growth past timepoints (Fig. 3 – unknown; Fig. 4 – 40 hrs, Fig. 7 – 40 hrs) is actually bacteriophage resistant host. The same types of growth curves can be obtained by performing low MOI infection assays. Culture must be re-seeded with phage and show no detriment with respect to uninfected controls in order to demonstrate that these are really BIM. Showing changes in CRISPR sequences or the acquisition of a resistance gene (i.e., on a plasmid or within the host genome) may also serve as added verification of bacteriophage resistance. In Figure 6, the authors could strengthen the manuscript by showing how the putative data set outcomes and interpretation differ when ISC, \square max, or phage score versus CI is used.

Thank you for your points and valuable suggestion. We have already corrected our terminology regarding “BIM”, since the focus of this study, as previously mentioned, is not to prove whether the increase in OD magnitudes is due to real BIM, or the epigenetic BIM. The focus and objective of this study is presented in a clearer way as it is to introduce a method which can compensate one of the clear errors made with current methods. The strength of the CI method is the assessment and distinguishing the position in growth curve where the highest cell density exists (like the considerable bacterial regrowth in some cases in presence of phages). Therefore, this unique property of CI enables researchers to evaluate the mutations like BIMs by this index, attributing the correct values to such curves. This characteristic of CI is not found in other available methods.

32. The centroid index (CI) described may have some utility; however, it needs to be clearly coupled to biology. Specifically, at any time point along the host growth curve, a measure of how many live cells is (i.e., cell density) determined using absorbance spectroscopy (i.e., optical density) as a proxy. A change from one time point to another show the net change in the number of new cells generated (e.g., via cell division) minus the number of cells lost (i.e., cell death) that occurred during t_2-t_1 . So, a concise understanding of slope-based measures between two time points (e.g., \square max) and area-under-the-curve (AUC) is provided in a physiological context. It is also easy to determine detriment or facilitation of host growth using these measures since a

difference can be calculated. The authors should expound upon what the centroid represents biologically. This is not clear in the manuscript. Furthermore, changes in the centroid from a control trial (e.g., uninfected host) versus an experimental trial (e.g., phage-infected) are clear if the difference occurs along a single axis (e.g., y-axis shift). The biological significance of the case where there is both an x-axis and y-axis shift is not clearly articulated.

Thank you for emphasizing on the biological significance of CI method. This section is added to the main text of the manuscript, and we have already elucidated the significance of CI method in justification of the biological dynamics of the bacterial growth curve.

33. It also seems that the authors have automatically assumed that what they are seeing in the psychrophilic phage system is lytic. The growth curves that presented are either non-lytic (i.e., blebbing/budding) or very low MOI traces where deep declines in culture are not occurring. The manuscript would benefit from showing a spot-on-lawn or “plaque” assay as well as any protein or mRNA-based evidence of a lytic replication cycle.

Thank you for your point, but in fact. It should be noted that our purpose in this manuscript is to bring up an index which can determine the problems in growth curves on the way of lytic infection/replication caused by a phage for phage therapy purposes. Having MOIs lower than the standard thresholds will not yield a successful bacterial reduction, and we think that this point could be also considered a problem. All in all, to be in the context of this presented study, we think that one should focus on the methodology of the suggested index, in addition to the strength it brings up compared to available methods.

34. Overall, the information presented in the manuscript is promising; however, the write-up needs clarification in several areas and a more detailed description of biological relevance of the centroid and changes in centroid. It is recommended that careful language regarding the utility of CI to multi-phage/single host interactions and interpretation be included in the manuscript.

We hope that our new version of the manuscript will be to your satisfaction. We added new analyzes and refocused the manuscript on the presentation of the method and its advantages, without seeking to overinterpret the possible reasons behind the fluctuations in growth curves to avoid distracting the reader from the most important aspect: the relevance of our tool to allow, among other things, an analysis of situations that are not well managed by other existing tools.

REVIEWERS' COMMENTS:

Reviewer #1 (Remarks to the Author):

Dear Authors,

I read carefully once again and found it had been improved a lot. Certainly, in the current version the manuscript is much more clear, on how the centroid method is different from AUC such as VI or PS parameters. This was misleading in the previous form in my opinion. Thank you very much for the additional examples, which support your findings and bring more light to this method.

I had other concerns, but I've found additional answers in the rebuttal letter sent to the other reviewers. Therefore, I can accept the author's response to my concerns. I suggest the Editor to accept the manuscript in its present form.

Best regards,

Reviewer #2 (Remarks to the Author):

The manuscript is improved after authors included comments of the reviewers. I'm satisfied how they answered to comments, and comments of other reviewers

Reviewer #3 (Remarks to the Author):

No further comments